# Code Integrity and Confidentiality: An Active Data Approach for Active and Healthy Ageing

**DOI:** 10.3390/s23104794

**Published:** 2023-05-16

**Authors:** Egor Litvinov, Henry Llumiguano, Maria J. Santofimia, Xavier del Toro, Felix J. Villanueva, Pedro Rocha

**Affiliations:** 1Nokia Corporation, 90620 Oulu, Finland; egor.litvinov@nokia.com; 2Department of Technology and Information Systems, University of Castilla-La Mancha, 13001 Ciudad Real, Spain; mariajose.santofimia@uclm.es (M.J.S.); xavier.deltoro@uclm.es (X.d.T.); felix.villanueva@uclm.es (F.J.V.); 3CINTESIS@RISE, Department of Behavioural Sciences, Abel Salazar Biomedical Sciences Institute (ICBAS), University of Porto, 4099-002 Porto, Portugal; parocha@icbas.up.pt

**Keywords:** code integrity, Active Data, code confidentiality

## Abstract

Internet of Things cybersecurity is gaining attention as the number of devices installed in IoT environments is exponentially increasing while the number of attacks successfully addressed to these devices are also proliferating. Security concerns have, however, been mainly addressed to service availability and information integrity and confidentiality. Code integrity, on the other hand, is not receiving proper attention, mainly because of the limited resources of these devices, thus preventing the implementation of advanced protection mechanisms. This situation calls for further research on how traditional mechanisms for code integrity can be adapted to IoT devices. This work presents a mechanism for code integrity in IoT devices based on a virtual-machine approach. A proof-of-concept virtual machine is presented, specially designed for providing code integrity during firmware updates. The proposed approach has been experimentally validated in terms of resource consumption among the most-widespread micro-controller units. The obtained results demonstrate the feasibility of this robust mechanism for code integrity.

## 1. Introduction

The Internet of Things (IoT) paradigm is one of the major leverages for the achievement of real smart spaces (the previously so-called *domotics*). Although the convenience of having a smart home can be appealing, the truth is that this field is now, yet again, gaining interest (after some decades of silence) as this could be an enabler for the active and healthy ageing domain. Smart homes could help older people live independently for longer, by relying on a set of smart sensors and actuators that monitor and supervise health and safety parameters, intervening when needed. While the price of IoT devices is decreasing, their functionality and capabilities are increasing. Smart homes are a reality as the technological readiness of electronic devices is advanced enough for the different applications that can be envisioned in such scenarios.

Within the landscape of the IoT, the sheer volume of connected devices, the vast amounts of data communicated between these devices and to back end systems, as well as the storage and analysis of data in the back end mean that security and privacy are critical in the IoT. For security, the IoT environment presents a major challenge: IoT devices typically have limited memory and compute power, and therefore, complex security features are not always possible. The volume of devices presents significant key distribution and key management issues, and schemes that facilitate scalability within a dynamic environment are required. Much of the data within the IoT environment contain Personal Identifiable Information (PII) and/or sensitive information, and therefore, privacy and access to information is also a key concern. To address end-to-end security requirements, security controls for confidentiality, integrity, and authentication must be flexible and practical, as well as enabling advanced features such as fine-grained access control.

Current industry standard mechanisms to achieve confidentiality, integrity, and authentication, such as symmetric key management systems and Public Key Infrastructures (PKIs), have serious shortcomings for large-scale and dynamic environments such as the IoT. Such schemes do not provide the required flexibility in dynamic environments. Symmetric key management systems do not scale well and need an Internet connection, and PKIs impose significant overheads, making deployments in resource-constrained environments complicated and often impractical. New practical and efficient encryption and digital signature schemes are required in IoT deployments to meet the demanded security requirements and give users control over the disclosure of private information.

Artificial Intelligence is also playing a relevant role in such a scenario as information gathered by sensors is generally translated into actions or events that describe ongoing situations in the monitored space. Smart actuators are employed, based on that interpretation, to intervene when needed. The work in [1] presents an architecture specifically devised to support actuation and smart behaviour capabilities in smart home environments.

The main idea of the smart home paradigm, therefore, relies on the device capability to reconfigure, at run-time, its functionality. Concerns about code security, in terms of confidentiality, integrity, availability, and authenticity, are gaining attention, especially after the effects of the Mirai attack. Despite the effects that a security breach could have on third parties, such as what happened to DynDNS during the Mirai attack [2], the real concern is the impact these can have on physical, emotional, and mental health. IoT devices for active and healthy ageing encompass a broad variety of electronic devices, among which we could name those monitoring health parameters such as heart rate, blood pressure, oximeters, or glucometers, just to name some of the most-popular ones. There are other devices with connectivity such as cardiac implants (pacemakers and defibrillators) or insulin pumps that are directly actuating over the individual’s health. The impact of a security breach in such devices could be catastrophic.

A medical device is a specific category of electronic devices that, when provided with connectivity capabilities, can also be labelled as an IoT device. Security mechanisms for such types of devices are stringent. However, as described in [3], attacks on cardiac implants have already been released, and the potential consequences of such actions definitely put people’s lives at risk. The articles in [4,5] describe attacks against insulin pumps. Both attacks demonstrate that remote control could be gained over the insulin pump, therefore re-programming the insulin supply. Similarly, the work in [6] describes the vulnerability found in a firmware update of a list of cardiac implants, subject to battery drain and crash attacks.

There are some other devices that do not fall into the medical category, but they could indeed put individual’s health in danger. In fact, the work in [7] analysed the interplay between robots, cybersecurity, and safety. The paper focuses on care robots, but the findings could be extended to any other cyber–physical system, as those supported on IoT devices. Like robots, devices deployed in a smart home environment for assisted living interact with older people. Mental health and emotional well-being are also major risks as pointed out by Fosch-Villaronga and Mahler. Devices such as speakers could be, a priori, thought to be harmless, in terms of risks to physical health, as they are not exerting a direct action over the physical world; however, risks could be experienced in the psychological dimension.

On the other hand, from the perspective of existing works from the literature focused on evaluating the security level or providing increased security mechanisms for IoT healthcare devices, we can mention the work in [8], which focuses on how to gather data from IoT devices in a secure manner. For this purpose, it proposes a novel encryption method called LRO-S to secure sensitive patient data and improve the safety and adaptability of medical professionals’ access to Cloud-based patient-sensitive data. The work in [9] focused on enabling penetration testing techniques in healthcare IoT devices, in an automatic and systematic manner. Finally, the work in [10] provides a rigorous and updated state-of-the-art revision of the available mechanisms for healthcare IoT devices. It is relevant to note that this work highlights the fact that, although the use of cryptographic keys is widespread in security architectures, there is limited research on creating, managing, and moving keys in resource-constrained environments. In this sense, our work contributes to this specific challenge as a novel and fit-for-purpose mechanism for low-resource devices is proposed, to ensure the secure communication of code updates.

This work focuses on a very specific asset of a computer system, its executable code. This work is, therefore, intended to ensure the security of the executable code, encompassing all aspects of security: code confidentiality, integrity, availability, and authenticity. Code (or application) security is a largely studied field with solid principles [11,12], guidelines [13,14], and standards [15,16,17], mainly for systems running Operating Systems. A few achievements can be named in this regard for Micro-Controller Units (MCUs) normally lacking operating systems or running firmware. However, the risks posed by code security breaches are among the most-challenging ones, as they can exert direct harm on the physical or psychological health of individuals. In this sense, advances in active and healthy ageing, mainly supported by IoT devices, call for a strong mechanism to ensure code security.

Inspired by the principles of Software-Defined Systems, the approach proposed here for code security resorts to the Active Data technology for improving the level of security during the communication (mainly involving code updates) between gateways or Fog computing devices and IoT nodes. The Active Data programming model has been successfully employed for addressing different issues such as described in Section 2.1.2. In security, the Active Data approach has been mainly employed for privacy. The work in [8] proposed the use of a software construct known as Active Bundle as a means to protect sensitive data from unauthorised disclosure and dissemination. An Active Bundle is comprised of the sensitive data, metadata, and a virtual machine. The role of the virtual machine is to validate the integrity of the Active Bundle itself. The focus of the proposed solution is on data privacy, and despite providing support for integrity, this is in terms of the Active Bundle integrity. Moreover, authenticity is not directly addressed.

The potential that Active Data could have for IoT ecosystems is currently being analysed from different perspectives [18]; however, once again, this is being solely applied to privacy. Furthermore, the proposed mechanism is expected to run in the Fog node, mainly due to the limited computational resources of edge nodes (mainly sensors and actuators). The work proposed here extends the application field of Active Data to support code confidentiality, integrity, and authenticity. The proposed approach will be useful for preventing different attacks against the confidentiality, integrity, availability, and authenticity of both IoT data and software, such as Denial of Service caused by a software crash or stealthy false data injection.

The scientific contribution of this work lies in proposing a novel approach to ensure code integrity in low-resource devices, by leveraging the concept of Active Data. This approach represents a breakthrough in the field of computer security, as it provides a resource-constrained mechanism for secure firmware update. While the idea of Active Data has been previously proposed for data privacy purposes, this is the first proposal to employ it for code integrity in low-resource devices. By using Active Data structures to embed the code update and its verification, the proposed approach addresses the limitations of conventional techniques that rely on cryptographic primitives, such as the AES and RSA. Moreover, the proposed approach does not rely on specific hardware requirements or additional memory, making it an attractive solution for low-resource devices.

The paper is organized as follows. First, the state-of-the-art is analysed from three main perspectives: (1) security principles that support this work; (2) security from the perspective of the IoT; (3) The Active Data concept. Then, Section 3 describes the proposed approach for code integrity in low-resource devices, with a specific focus on devices applied in the field of healthy and active ageing. Section 4 presents the experimental validation of the proposed approach, using for that purpose some of the most-commonly used micro-controller units in the field. Finally, Section 5 presents the main conclusions of this work.

## 2. Previous Works

### 2.1. About Basic IoT Security Concepts

The three key elements of any security scheme is authenticity, integrity, and confidentiality. Authenticity in the IoT means confirming the source of any message received by the IoT device; integrity consists of confirming that the received message has not been modified since it was sent by the authorized source; confidentiality ensures that unauthorized entities cannot understand the content of the messages even if they can gain access to them. Directly related to the security scheme is the authorization of each entity and the privileges of those entities and how to manage those privileges.

A message is any type of communication between two entities including a data message with a payload, a firmware update, or a new configuration setup. In the IoT domain, the most-challenging scenario involves a low-cost low-footprint IoT device communicating with a gateway.

Integrity is directly related to the hash functions used to generate message digest. A hash function is a mathematical function that takes a block of original data, and it calculates a string of a fixed length with a summary value (the so-called message digest). Any change in the original data leads to the generation of a different digest, and for that reason, this mechanism is normally employed to detect changes in the original data (e.g., due to an error or a security attack). A good hash function makes it almost impossible to obtain the original data from the output (preimage-resistant), and it makes it impossible to find two different data blocks with the same summary (second preimage-resistant and Collision Resistant Hash Function).

If a well-known hash function is used to generate the message digest, for example cyclic redundancy code hash functions, the message digest is used to detect any variance/error in the original data. If in the generation of that message digest, a private/public key is used, a digital signature for that message is also obtained, therefore providing it with authenticity. In other words, the message digest also works as a digital signature, and therefore, it also provides information about who generated that message.

Ensuring authenticity and integrity for scenarios involving the update of firmware and/or new configuration setups is a challenge in low-cost Micro-Controller Units (MCUs). However, in any IoT project, the adopted security scheme is a trade-off between the overall required security (risk of being compromised and time to recover), the cost of the adopted solution, and the performance of the adopted solution. The performance on the IoT device side is the time to encrypt and decrypt a message and to authenticate a firmware/configuration update (including the impact on power consumption).

Symmetric cryptography relays on a secret key stored in the MCU and in the maintenance centre, employed to cipher and decipher any message transmitted from the maintenance centre to the MCU and the other way around. There are several algorithms that can be used for implementing symmetric cryptography schemes. The mayor drawback, nonetheless, is that, when the MCU is physically accessible, the overall scheme can be compromised if the key is stolen. Furthermore, most secure schemes require high computational capabilities, which exceeds most low-cost MCUs.

The Advanced Encryption Standard (AES) is a symmetric block ciphering approach that takes blocks of a configurable size (128, 192, or 256 bits), and using a configurable number of rounds (10, 12, or 14 bits), it ciphers the message using four operations. The AES is a standard with a considerable level of security, but it also has strong computation requirements.

Public Key Cryptography (PKC) or the asymmetric cryptography infrastructure relies on two pairs of related keys (public and private) in such way that a message ciphered with the public key can only be deciphered with the private key. Under this scheme, for example, the server signs or ciphers any message with the public key of the embedded node in such a way that the embedded device verifies or deciphers messages using the private key. The mathematical properties of asymmetric cryptography guarantee that security is not compromised, even when the public key is known. The major drawback of this approach is the computing requirements for verifying the authenticity of a message since most secure implementations (e.g., the FIPS 186 Elliptic Curve Digital Signature Algorithm) exceed the computing capabilities of most low-cost MCUs.

One approach to deal with the computing limitations is to use specific hardware modules designed for security functions (cipher, authentication, etc.). These hardware modules are too expensive to be used for some IoT devices. To overcome this limitation, lightweight implementations are being provided to address security on IoT devices. Currently, the NIST security agency has an ongoing process to select the next generation of lightweight cryptography algorithms (https://csrc.nist.gov/projects/lightweight-cryptography/round-2-candidates, accessed on 12 March 2023). Nevertheless, lightweight cryptography still pursues a trade-off between computer requirements to achieve an acceptable performance and the obtained level of security.

#### 2.1.1. Security in IoT

The first approaches to the IoT were mainly based on Cloud solutions. The use of the Cloud offered a simple way (although not the most-efficient one) of addressing the heterogeneity (in terms of devices and networks) in the IoT. These approaches have many inconveniences such as: high latency, resource consumption, or low fault tolerance. All these inconveniences led to the proposal of a new computing approach known as Fog Computing [19] to provide compute, storage, and networking services between Edge Computing devices and Cloud Computing platforms. This definition does not clearly state what falls into the Edge Computing layer and what falls into the Fog Computing one. The work in [20] addresses this, and other related questions. From the security and privacy perspective, as stated in [20], the Fog Computing layer can provide more advanced security services for the things of the IoT (Edge devices), basically because they will be less constrained in terms of resources and capabilities. Moreover, Fog nodes can work as proxies or gateways, in which security mechanisms can be enforced. It is also common to refer to this layer as the *Edge Server Layer* [6], Thin Layer [21], or User Edge [22].

Nevertheless, security at the Edge Computing layer cannot be solely based on enforcing stringent security mechanisms at the Fog layer. In fact, the recent increase in attacks to Edge Computing platforms [23], being one of the most-widely known one the one articulated by the Mirai malware [24], is calling for more solid mechanisms. The lowest tier of the edge taxonomy is the one that, due to its low-resource capabilities, poses major security challenges. On the other hand, this is the tier that, when breached, can cause major damage, also in terms of physical harm to end users. This is the tier addressed by this proposal.

Two types of devices can be found at the lowest tier of the edge taxonomy: IoT devices and mobile devices [23]. This work focused on ensuring code confidentiality and integrity for IoT devices. These devices are normally lightweight electronic devices, most of them with a Micro-Controller Unit (MCU) based on a Cortex-M series, interconnected using wireless protocols, WiFi, Bluetooth, or ZigBee.

#### 2.1.2. Active Data

The concept of the Active Data structure was originally proposed by G. Andrews, D. Dobkin, and P. Downey [25], in opposition to classical data structures, considered to by passive software agents. The idea was formulated in the area of computer architectures, in 1981, as an innovative mechanism to address the new challenges brought about by the multiprocessors era. The Active Data concept was based on the idea of combining the data along with the operations to be run on them. This idea was opposite that of data flows in which data go through different processors in which different operations are being applied. In this case, the processors receive the data and the operations to be performed overcoming the memory bottleneck, as well as enabling a higher degree of parallelism.

Similarly, still in the field of the computer architecture, there is the idea of active message [26]. In this case, the purpose of the active message is to minimise the communication overhead in large-scale multiprocessors when using approaches based on message passing. The messages are self-contained structures.

The work in [27] resorted to the concept of Active Data as a means to address power consumption in Wireless Sensor Networks (WSNs). In this case, the idea of Active Data is extended to a local data area. Energy consumption is reduced during data dissemination because the data sink can obtain the data actively from the sensor node that will always be at just a one-hop distance.

The idea of Active Data has inspired many others, in which the objective is that of combining the data and the operations to be performed on those data. More recently, this idea has been applied in fields other than computer architectures or networks, as is the case for computer security. For example, the work in [8] presents the idea of an Active Bundle. This extends that of the Active Data because, in this case, the Active Bundle is a container with a payload of sensitive data and metadata, but also a virtual machine. The virtual machine, rather than being a general one, is made fit-for-purpose for the sensitive data and the metadata that accompany the virtual machine. This idea was proposed as a mechanism to protect the data privacy of sensitive information all over the data lifecycle (from the creation of the data, up to its eventual destruction).

In the field of computer security as well, Active Data have been applied to stealthy false data detection [28]. In this case, false data injection attacks are detected by implementing an approach that actively modifies the data (measures) before being transmitted through the network. On the other side of the network, data are turned into their original values. The data on the reception point are expected to meet a certain pattern. The data have been modified before their transmission to meet that pattern. False data can be easily identified when the expected pattern does not match. Furthermore, the computational overhead or message size does not affect the system performance.

The main reason why the Active Data idea is recurrently being applied to different fields, since its first proposal in 1981, is because it provides a constrained mechanism, in terms of resource consumption, for combining in a single resource both the data and the procedures to be performed on that data. Although Active Data have been proposed for computer security, the focus has been on data privacy. This is the first approach that employs the Active Data structure for code integrity in low-resource devices.

## 3. The Proposed Active Data Framework

The proposed active data framework is intended to provide a secure and trustworthy method for firmware updates in IoT scenarios. In this sense, the firmware update will rely on a set of elements that, jointly, will ensure the integrity, confidentiality, and authenticity of the firmware update.

These elements can be basically categorized as:The data: This is the code to be used to update the MCU firmware. Code confidentiality and integrity should be ensured, as well as authenticity in terms of being provided by an authentic source or, in other words, a source being authorised for performing software upgrades.The metadata: This information is employed for establishing a secure communication channel between the node sending the software update and the one receiving it.The virtual environment: This is an emulated environment that works as a sand box, in which the code authenticity, confidentiality, and integrity is verified and, eventually, deobfuscated to be successfully installed.

The following subsections will describe the details of these elements, independently of the MCU on which they will be deployed.

### 3.1. Description of the Data

The most general case in the IoT is to have an MCU node, controlled by a low-level software known as firmware or, in the most-advanced ones, by a real-time operating systems such as FreeRTOS [23]. This software is normally stored in the Flash memory, and most of them are loaded by another software known as the *bootloader*. The bootloader is commonly copied in the same Flash memory, from the first position, although there are other configurations as for the Arduino Zero board, in which case it is pre-recorded during the manufacturing process in the ROM memory of the chip.

The memory map of the device specifies the different types of memories and their sizes for a particular board. For example, Figure 1 outlines the memory layout for the Arduino Zero. The aforementioned Flash memory behaves as an instruction memory, whereas the SRAM behaves as a data memory, where the stack is commonly located. In any case, one of the aspects that needs to be determined is where instructions are to be located, because this will be the memory range that will be overwritten during the firmware update.

In this sense, any software update will replace the instructions located in the instruction memory, physically mapped in the case of the Arduino Zero to the Flash memory, and more specifically from memory position 0x00000000 to 0x2000000. The bootloader copies the new instructions into that region and launches the execution once the copy has finished. Nevertheless, the main drawback of this traditional process is that the bootloader performs the update without checking the integrity of the code, nor the authenticity of the source performing the upload. Furthermore, the code that it is to be uploaded is in plain machine code, so that it can be directly executed by the processor, but easy to disassemble and, therefore, not confidential.

Confidentiality is generally achieved by a using encryption schemes. Nonetheless, due to performance reasons, this is not an option unless the MCU is equipped with specific hardware for performing such operations. The use of such hardware will also increase the price of the MCU, so most of the boards employed in IoT scenarios will lack this capability. An alternative approach consists of performing some basics operations that lead to a message obfuscation, but the operations employed to do so are lighter than current ciphering approaches. Ideally, the obfuscating operation has to be easy to reverse and easy to compute in order to overcome the limitations of traditional ciphering schemes. The XOR function is an operation that complies with these two requirements. It is extensively used for this purpose as it is straightforwardly computed by most arithmetic-logic units, and its reversion consists of applying the same function over the obfuscated message. The idea proposed here to ensure code confidentiality is to obfuscate the instructions comprising the update by applying an XOR operation.

The XOR operations will be performed using a random number provided as part of the metadata and agreed among the two parties participating in the communication. The XOR operations will be performed by the virtual environment running on the two sides of the communication. The idea is to make the machine code go through a compilation process, during which the instructions are obfuscated by an XOR operation. On the other end, the virtual machine will perform the role of a code interpreter and will decode each of the instructions before copying them into the instruction memory.

### 3.2. Description of the Metadata

The virtual environment on the two sides of the communication needs to agree on a set of parameters so that the instructions comprising the update can be decoded and verified in terms of integrity and the authenticity of its source. These parameters comprise the metadata that will travel from one side to the other so that both of them can agree on the terms of the communications to ensure the confidentiality, integrity, and authenticity.

The proposed approach is referred to as an active data approach because the information required to process the software update comes along with the code update itself. In other words, the data required to ensure the confidentiality, integrity, and authenticity of the code update accompany the code update itself.

The data that comprise the metadata are the following:The public key of the node that sends the software update, referred to as *PK1*, standing for the public key of Node 1.A random number of 16, 32, or 64 bits in length (referred to as *r64* in the case of instructions of 64 bits). The random number size depends on the instruction size of the processor. This random number is used to generate the obfuscated instructions by performing the XOR operation as follows: “final_instruction” = “default_instruction” XOR “*r64* ”. The purpose of this operation is to make a reverse engineering process based on brute force more difficult. With this basic obfuscation, the operation codes will be different on every initialization of the virtual machine.A set of random numbers (*magic_numbers*) that will be used as the operand for the execution of a selected list of instructions. These numbers will be stored in the virtual registers *R0-R3*.A list of instructions, referred to as *list_2*, randomly selected from those comprising the Instruction Set Architecture (ISA) of the processor. These instructions will be executed using the *magic_numbers* stored in registers *R0-R3*. The values of registers *R0-R3* after having executed such instructions will be kept in *result_2*.

### 3.3. The Virtual Environment

The virtual environment performs the role of a virtual machine, which once initialized with the metadata, will ensure that both sides of the communication work over a common environment.

The side that starts the communication waits for the metadata provided by the node to be updated, and based on such metadata, it initializes its virtual environment. Once the communication is established and the virtual environment initialized, the obfuscated instructions are provided to the node to be update, which after its deobfuscation, will be provided to the bootloader, which will, eventually, copy them into the Flash memory and restart the processor to launch the updated firmware.

There are two stages in this process: (1) the communication establishment and (2) the code transference and installation. These two stages are described below.

#### 3.3.1. The Communication Establishment Protocol

The first stage of the firmware update consists of establishing a secure communication channel so that both nodes can be assured that they are authentic and that the code travels in a confidential manner and with its integrity being complete. During the establishment of such a secure channel, a metadata interchange is carried out that will support the later process of code obfuscation and deobfuscation. The establishment of such a secure channel follows the steps described in Figure 2.

The node (Node 1) that intends to update the firmware of the other node (Node 2) starts the request for the communication establishment by sending the initialization command, along with its public key (PK1). This public key needs to be verified with a trusted authority, to determine whether Node 1 is among the nodes that have been authorized to perform the update on the IoT nodes. This trusted authority runs on a known server, and its location is pre-configured in the virtual environment of the node. Normally, this will be a key server running on a Fog node or gateway. Once Node 2 verifies the identity and the permissions to perform the firmware updates of Node 1, it starts the initialization of its virtual environment. During the process, the metadata, behaving as the Active Data, will be generated and sent back to Node 1. More specifically, the following metadata will be computed:The random number employed to obfuscate the machine instructions (the length will depend on the length of the instructions for that ISA).Four random numbers, known as magic numbers, to be stored in registers *R0-R3*.A list of four instructions, randomly selected from the firmware update. These instructions will exclude those of flow control, and therefore, basically arithmetic instructions will be considered.

The random list of four instructions will be executed, using the values stored in registers R0-R3. Because only arithmetic instructions are considered, the result of executing such instructions will alter the original values stored in registers *R0-R3*. These new values are stored in a metadata data structure known as *result_2*.

The metadata calculated during the initialization phase is packed in a message, ciphered with the public key of Node 1 (*PK1*). This ensures the confidentiality of the metadata as only the node with the corresponding private key can decipher the message.

The transference can be carried out using any predefined protocol, such as WiFi, Zigbee, UART, etc. The only restriction is that Node 2 needs access to the certification authority, so whatever the transference protocol employed, the node running the key server has to be reachable.

Upon reception of the ciphered message, Node 1 uses its private key to decipher the message and extracts the random number *r64*, the *magic_number*, and the list of instructions (*list_2*) and uses this to initialize its virtual environment. After having stored the values of the *magic_numbers* in the corresponding registers *R0-R3* and following the random list of instructions of *list_2*, the result in registers *R0-R3* is stored in a data structure referred to as *result_1*, then sent back to Node 2, which will compare it with the previously calculated *result_2*. If both results match, then the initialization stage concludes and the connection is established.

It is important to note that the only cryptography operation performed by Node 2 is the ciphering of the message with the metadata needed to initialize the virtual environment. The rest of the operations intended to ensure confidentiality, authenticity, and integrity are the extent of the cryptography operations, which are supposed to have a high demand of computational resources.

#### 3.3.2. Code Update Transference and Installation

During the previous stage, Nodes 1 and 2 have established a secure channel for data transference, so during this second stage, the whole code update is transferred from Node 1 to Node 2, and afterwards, the update is installed, replacing the previous firmware.

The previous stage concludes with both nodes having the virtual environment initialized in the same way. Before transferring the code update from Node 1 to Node 2, Node 1 obfuscates all the instructions so that if the transference is tampered with, the code confidentiality can resist a brute force attack, as long as possible. Therefore, in the same way that, during the initialization process, the random instructions of *list_2* were obfuscated using an XOR operation with *r64*, the instructions comprising the firmware update will go through the same process. The process is described in Figure 3.

Once the code is deobfuscated in the target node (Node 2), the firmware update is ready to be installed. The instruction memory cannot be directly overwritten as the virtual machine is running on that same memory, and some precautions and mechanism should be in place in order to ensure that running instructions are not overwritten, leading the processor to an unstable state. On the other hand, the bootloader could be updated to include a new directive supporting the installation of the firmware update, while still running the previous version.

Generally, bootloaders employ a serial connection from which the firmware update is received, launched after an interruption (reset) triggers the bootloader. In order to enable the reception and installation of firmware updates from interfaces other than the serial one, the bootloader needs to be updated so that the instructions, once deobfuscated in the virtual environment, can be provided to the bootloader, in charge of eventually copying them into the instruction memory. Before delegating the control to the bootloader, Node 2 sends a message to Node 1 to notify it that the instructions have been successfully deobfuscated. Then, the virtual machine is exited and the bootloader takes the control of the whole process, copying the instructions from the memory region where the virtual environment has stored the deobfuscation instructions, to the Flash memory where the processor will start the execution after the reset. Node 1, after receiving the message of the successful update, will also exist the virtual environment.

## 4. Experimental Validation

Figure 4 outlines the organisation of the project for the proof of concept implementation.

The FreeRTOS section contains all the necessary files required by the operating system. This is the case for FreeRTOS, but it can easily be any other operating system specifically designed for embedded systems. Furthermore, it is important to note that the proposed infrastructure can also work in a *bare metal* mode, without an operating system, as is the case for the validation tests run here. For the case in which FreeRTOS is employed, there is the project code section, normally in C or C++, which is devoted to performing some operation, normally based on the captured data or intended to perform an action under certain circumstances. Finally, ADVM provides the virtual environment for the active directory framework proposed here. There are three main files providing the implementation for this virtual environment:The header file *advm.h* contains the definition of the different functions in charge of the virtual environment initialization and the function supporting the secure communication between two nodes.The implementation of the functions defined in the header file are provided in *advm.c*.

To validate the proposed framework, two specific MCUs have been selected, as representative of the most-widespread embedded architectures: the Arduino Zero and HiFive1 rev B boards. The proposed active data framework was implemented and executed, and some metrics were collected to evaluate the burden associated as a result of the proposed methodology:Space in memory occupied by the virtual environment infrastructure, the decode function, and the main function.Space, in percentage with respect to the available memory in the Flash memory.Time to perform the setup and main functions.

Table 1 summarises the main features of the two boards used for the experimental validation of the proposed infrastructure.

Code Listings 1 and 2 provide the details of the functions that were evaluated, both in terms of memory space and execution time.

**Listing 1.** C code for the setup function.
**#include** <Arduino.h>
**#include** <stdio.h>
**#include** <stdlib.h>
**#include** <advm.h>

**void** setup(){
   **volatile** **long** **int** t0, tf;
   t0=micros();
   uint8_t answer_vm;
   uint8_t data_cmd[] = { PUSH, 33,
          PUSH, 35, 
          ADD};
   vm_env(VM_INIT, NULL, &answer_vm);
   vm_env(VM_MSG_RX, &data_cmd, &answer_vm);
   tf=micros();
   **return** 0;
}


**Listing 2.** C code for the main function.
**#include** <stdio.h>
**#include** <stdlib.h>
**#include** <advm.h>
**#include** <metal/time.h>
**int** main(){
   **struct** timeval begin, end;
   metal_gettimeofday(&begin, 0);
   uint8_t answer_vm;
   uint8_t data_cmd[] = {PUSH, 33,
         PUSH, 35,
         ADD};
   vm_env(VM_INIT, NULL, &answer_vm);
   vm_env(VM_MSG_RX, &data_cmd, &answer_vm);
   metal_gettimeofday(&end, 0);
   **return** 0;


Table 2 summarizes the results obtained for the two boards considered in the experiment. It has to be highlighted that the impact on the memory space is nearly almost negligible. The impact on the time performance is also very low, as, for the Arduino Zero, the framework requires less than 200 cycles, whereas for the HiFive1, it takes roughly 3000 cycles. It is important to note that the code in the HiFive1 is run from the Flash memory, which is slower than using the cache memory (ITIM).

The classic approach of generating a random key (DES) to encrypt the recipient’s public key and using it to XOR every instruction of the code update is susceptible to known plaintext attacks. A malicious attacker who intercepts both the encrypted key and the encrypted code can deduce the key by XORing the two and comparing the result to the unencrypted code. On the other hand, the proposed approach involves using random arithmetic instructions and magic numbers to obfuscate the code, making it much more challenging for an attacker to deduce the original code, even if he/she has access to the encrypted code and initialization metadata. Because the three random arithmetic instructions can be updated periodically, this helps to prevent attacks that rely on fixed encryption algorithms. Additionally, this method offers greater control over which parts of the code are obfuscated, as only arithmetic instructions are used for the metadata generation. Moreover, incorporating a trusted authority to verify Node 1’s identity and permissions adds an extra layer of security to the process, ensuring that only authorized nodes can perform firmware updates.

The advantage of the proposed approach to initialize the virtual machine operations lies in the additional layer of security it provides to the firmware update process. Furthermore, the described approach allows for a more efficient use of computational resources, as the obfuscation is performed within the virtual environment, which is specifically designed for this purpose. In the classic approach, the XOR operation would have to be performed on every instruction, which could be computationally expensive, especially for larger firmware updates. Whereas in the approach proposed here, code confidentiality and integrity can be sacrificed, when no other choice is available, code authenticity has already been guaranteed at the beginning of the code transfer. Furthermore, less-relevant pieces of code, such as those containingtext (.TEXT section), can be the ones selected to be transferred in plaintext, mitigating the effects of a code tamper. This approach allows for more fine-grained control over which parts of the code are obfuscated. The potential is, therefore, in the flexibility it offers for very low-resource devices that need to be updated, providing them with the maximum level of security.

This approach also presents some weaknesses as using a constant key to encrypt code by XORing can lead to a vulnerable solution. This is because it creates an ECB-like solution that is susceptible to frequency analysis. The Electronic Codebook (ECB) is a mode of operation for a block cipher, in which the message is divided into blocks of a fixed size, and each block is encrypted independently using the same key. The drawback of ECB mode is that identical plaintext blocks are encrypted into identical ciphertext blocks, which can reveal patterns in the data and potentially make it vulnerable to attacks. Using Cipher Block Chaining (CBC) or a similar mode of operation would be a better solution because it provides better confidentiality and integrity protection. In CBC mode, each block of plaintext is XORed with the previous ciphertext block before being encrypted. This means that, even if the same plaintext block is encrypted multiple times, it will produce a different ciphertext block each time, making it resistant to frequency analysis. Nonetheless, the encryption process requires a cryptographic key, an Initialization Vector (IV), and a block cipher algorithm, such as the AES. The IV is XORed with the first block of plaintext before being encrypted with the cryptographic key and block cipher algorithm. The resulting ciphertext is then used as the IV for the encryption of the next block of plaintext, and so on. This means that the encryption process using CBC mode requires additional computational overhead to generate a random IV for each block of plaintext. In contrast, using a constant key to encrypt code by XORing eliminates the need for an IV and reduces the computational overhead required for encryption. Furthermore, using a constant key for XOR encryption also allows for faster decryption, since the same key can be used for decryption as well. In contrast, with CBC mode, the decryption process requires the use of the same cryptographic key and the same IV used for encryption, adding additional computational complexity to the decryption process.

A comparative analysis of the DES algorithm implemented in the two boards employed for evaluation purposes is presented in Table 3. The algorithm implementation is based on https://arduino-projects4u.com/software/des022.ino (accessed on 12 March 2023). It can be observed that the code footprint increases with the DES implementation. Regarding execution time, it significantly increases even when decrypting only 64 bits, which is the basic fragment of information used in the DES algorithm. The overall time will therefore increase depending on the size of the firmware to be updated. The firmware size of the basic implementation, without any additional code is also provided in Table 3.

## 5. Conclusions and Future Work

The active and healthy ageing paradigm is gaining attention as this provides an alternative for institutionalised care. The deployment of technological solutions that monitor older adults while being at home and on the move, both in terms of behaviour and health, provides a secure alternative for older adults to live independently as long as possible. In terms of technology, teleassistance and smart homes are now a reality. The problem, nonetheless, is that there is little attention paid to those devices that populate homes, most of them low-cost and low-resource. The appearance of vulnerabilities and the obligation in the EU market to keep devices updated open the door to attacks covered as a stealthy firmware update.

This paper presented an approach for the firmware update of devices that lack the resources to run ciphering approaches. The proposal was based on the concept of Active Data. A structure was built that combines symmetric and asymmetric ciphering to establish a secure channel with the device to be updated. The proposal was validated with some of the most-common micro-controller units, widely employed in active and healthy ageing environments. The results demonstrated that the load in terms of memory and time is affordable, therefore achieving a reasonable trade-off between resource consumption and the security level achieved.

Future works should address the situation in which the size of the update firmware is larger than the available space in the SRAM. In this case, one approach to optimize the firmware update is to split the update into smaller chunks that can be loaded into the SRAM and executed sequentially. This approach can be combined with a verification step for each chunk of code that is loaded into the SRAM, to ensure that it has been loaded correctly and without errors. A means will also need to be provided to developers so that the firmware updates can be carefully designed and tested to ensure that it is able to work within the constraints of the available SRAM space and that it is able to update the firmware without causing any adverse effects on the operation of the device.

## Figures and Tables

**Figure 1 sensors-23-04794-f001:**
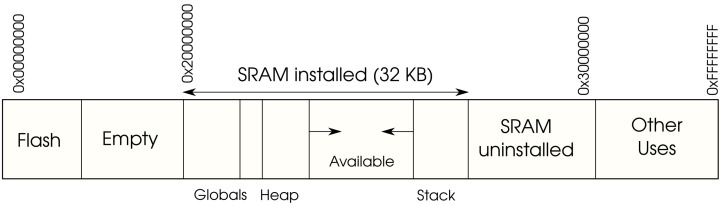
Memory organization of a specific MCU (Arduino Zero).

**Figure 2 sensors-23-04794-f002:**
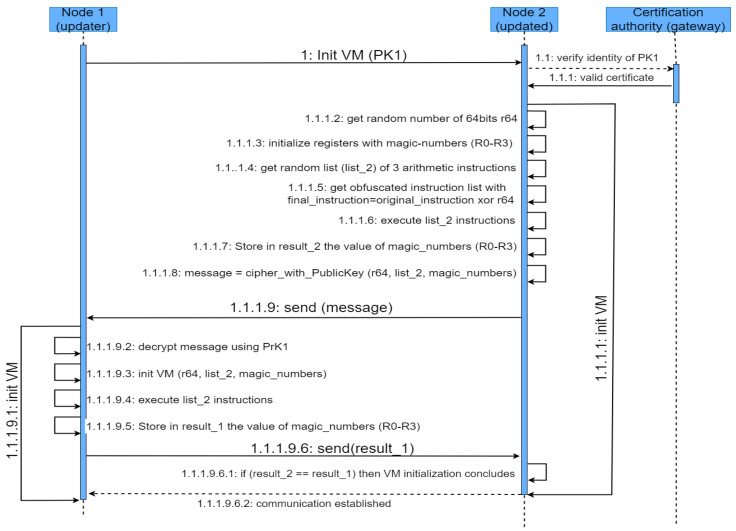
Overview of the proposed protocol for the establishment of a secure communication.

**Figure 3 sensors-23-04794-f003:**
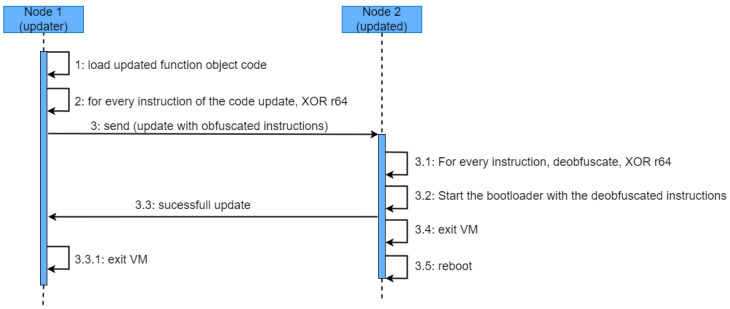
Overview of the transference and installation protocol.

**Figure 4 sensors-23-04794-f004:**
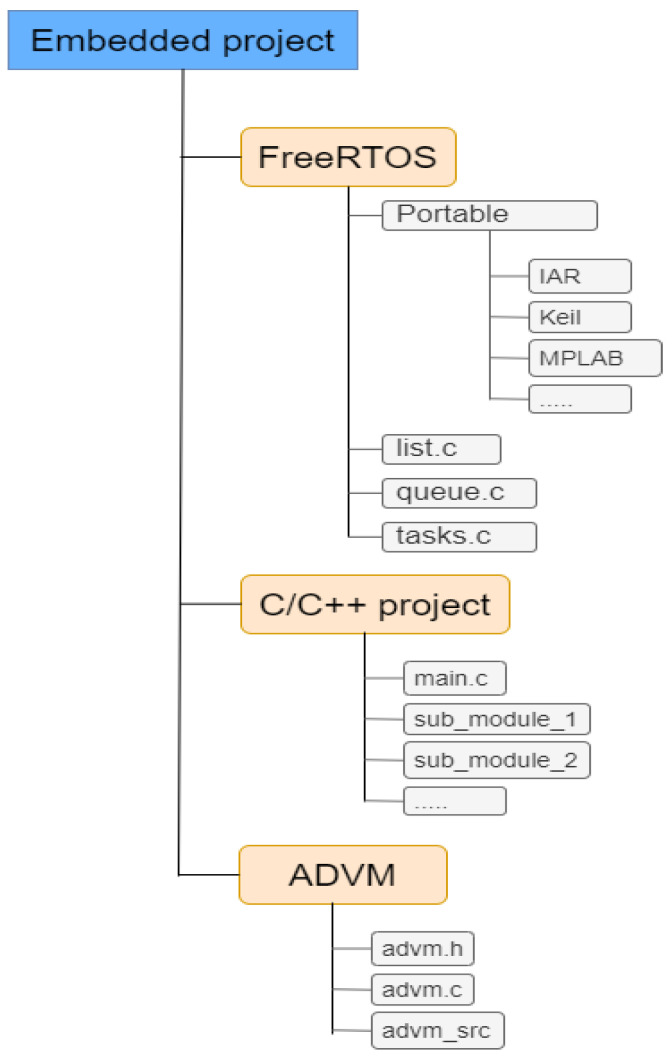
Typical organisation of an embedded project.

**Table 1 sensors-23-04794-t001:** Main features of the two boards used in the evaluation.

Board	Arduino Zero	HiFive1 Rev B
MCU	32-bit Atmel SAMD21G18A	32-bit SiFive FE310-G002
Processor	ARM Cortex M0+	SiFive Freedom E31
Frequency	46 MHz	320 MHz
Architecture	Armv6-M	RV32IMAC
Flash Memory	256 KB	4 MB
RAM Memory	32 KB	16 KB

**Table 2 sensors-23-04794-t002:** Obtained results for the two boards used in the evaluation with the proposed solution.

Board	Arduino Zero	HiFive1 Rev B
vm_env function	88 bytes	114 bytes
decode_vm function	68 bytes	96 bytes
main function	64 bytes	68 bytes
Total code footprint	220 bytes	274 bytes
% of Flash memory	0.00083%	0.000065%
setup/main execution time function calls	37 µs	9379 µs

**Table 3 sensors-23-04794-t003:** Obtained results for the two boards used in the evaluation with the DES algorithm.

Board	Arduino Zero	HiFive1 Rev B
des_dec function	588 bytes	728 bytes
des_f function	200 bytes	202 bytes
changeendian32 function	60 bytes	76 bytes
permute function	96 bytes	112 bytes
main function	52 bytes	42 bytes
total code footprint	996 bytes	1160 bytes
% of Flash memory	0.0038%	0.00027%
setup/main execution time function calls note: only 64 bit decrypted	11,941 µs	116,420 µs
firmware size (.bin, .hex)	17 KB	68 KB

## Data Availability

Not applicable.

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
