# Peer review of "Code Integrity and Confidentiality: An Active Data Approach for Active and Healthy Ageing"

_sensors, 2023, doi:10.3390/s23104794_

Round 1

Reviewer 1 Report

The manuscript deals with the important problem of security of IoT devices especially in the context of secure firmware update. Proposed approach provides code integrity and confidentiality during firmware upgrading using active-data approach and is especially devoted to IoT devices with limited computational capabilities. The presented solution seems to be working correctly but in my opinion it is unnecessary complicated and provided level of security is rather low. Especially I would like to ask authors to address the following issues:

1. Since it is assumed that there is access to the trusted PKI why not to use a classic approach like:

- Node 2 generates random Key, encrypts it with public PK1 key and sends it to the Node 1, then this key is used to XOR every instruction of the sent code update.

- Node 2 can authenticate Node 1 using standard challenge-response technique.

2. What is the advantage of described init VM operations (e.g. getting 3 random arithmetic instructions) in comparison with mentioned above classic approach?

3. “Encrypting” transferred code by XORing it with the constant (during one session) key creates ECB-like solution that is not considered as a secure (e.g. because subsequent encryptions of the same plaintexts result in the same ciphertexts). In fact is a substitution cipher which is susceptible to frequency analysis. Maybe using CBC (or similar) mode would be a better solution (and still with low computational complexity).

4. If the size of the updated code is limited to the SRAM memory size? In case of Arduino Zero module SRAM size is much smaller than flash size.

5. In section “4. Experimental validation” it is worth to compare effectiveness of the proposed obfuscating (by XOR) method with a “real” cipher like DES.

I also have some editorial comments:

- What is the purpose of Figure 1 since it is even not referenced in the text?

- In Figure 2 in step 1.1.1.8 rather the Public Key should be used.

- In Figure 3 descriptions are difficult to read because of the very small font.

Regarding language I noticed only few typo mistakes:

- line 166: should be asymmetric instead of asimetric

- line 282: should be Flash instead of Flahs

- line 416: “mechanims”

- line 449: lack some word after “two specific MCU have been”

- line 497: “cicles”

Author Response

Dear reviewer

Reviewer 2 Report

The authors present a code integrity mechanism in IoT devices based on a virtual machine approach to ensure code integrity during firmware updates. They experimentally validated the proposed approach in terms of resource consumption among the most widespread microcontroller units. The scientific contribution of this work is well justified. The paper itself represents an interesting contribution, relevant to the scope of the journal and valuable for field of theoretical aspects related to software engineering process in context of IoT. Despite the uniqueness and solid outcome including the list of challenges, some remarks should be adressed to impove the quality of the paper:

-       -    Add a paragraph at the end of the introduction that clearly describes the scientific contribution of this work.

-         -  Figures 3 and 4 require quality improvement.

-     -      Add recent rederences (for example those published in 2022).

Author Response

Dear reviewer

Round 2

Reviewer 1 Report

Regarding the Response 1 and the Response 2:

I agree that just XORing with the Key is vulnerable to known-plaintext attack but also in your solution with VM operations (e.g. getting 3 random arithmetic instructions) the transferred code protection is still based on the XORing with the Key (in this case r64 value). In order to perform known-plaintext attack successfully you need to bruteforce the Key, and its length (64 bits) provides rather a sufficient level of protection.

I can’t agree with your answer that “In the classic approach, the XOR operation would have to be performed on every instruction, which could be computationally expensive, especially for larger firmware updates
because in your solution (if I understand it correctly) the XOR operation must be performed also on every instruction (it is even clearly stated in Figure 3).

In Response 3 you wrote: “This means that the encryption process using CBC mode requires additional computational overhead to generate a random IV for each block of plaintext”.
In fact in CBC random IV is generated only once and then used in the whole chain. It seems to me that in this sentence a mental shortcut was used and it is more about the computational effort associated with XOR operations between subsequent plaintext blocks and previous ciphertexts (which is rather low).

Regarding editorial, I think that font size in Figure 2 is still too small.

Author Response

Dear reviewer
